# A Recipe for Charge Density Prediction

**Xiang Fu**[1]* **Andrew Rosen**[2,3] **Kyle Bystrom**[4] **Rui Wang**[1] **Albert Musaelian**[4]

**Boris Kozinsky**[4,5] **Tess Smidt**[1] **Tommi Jaakkola**[1]

[1]Massachusetts Institute of Technology

[2]UC Berkeley [3]Lawrence Berkeley National Laboratory

[4]Harvard John A. Paulson School Of Engineering and Applied Sciences

[5]Robert Bosch Research and Technology Center

## Abstract

In density functional theory, the charge density is the core attribute of atomic systems from which all chemical properties can be derived. Machine learning methods are promising as a means of significantly accelerating charge density predictions, yet existing approaches either lack accuracy or scalability. We propose a recipe that can achieve both. In particular, we identify three key ingredients: (1) representing the charge density with atomic and virtual orbitals (spherical fields centered at atom/virtual coordinates); (2) using expressive and learnable orbital basis sets (basis functions for the spherical fields); and (3) using a high-capacity equivariant neural network architecture. Our method achieves state-of-the-art accuracy while being more than an order of magnitude faster than existing methods. Furthermore, our method enables flexible efficiency–accuracy trade-offs by adjusting the model and/or basis set sizes.

## 1 Introduction

Density functional theory (DFT) is a computational quantum chemistry method that has enabled countless advancements in the chemical sciences by providing a tractable means to calculate the electronic structure of molecules and materials [1]. The central concept in DFT is the charge density, a fundamental quantity from which all derivable ground-state physicochemical properties of a system, such as energy and forces, can, in principle, be derived. The most widely used Kohn–Sham formalism [2] of DFT offers a reasonable balance between accuracy and computational efficiency among conventional DFT workflows. However, it still scales with a complexity of roughly $O(N_e^3)$ where $N_e$ is the number of electrons, rendering it computationally expensive and limiting its viability for both large-scale systems and long-timescale *ab initio* molecular dynamics simulations.

In DFT, the solution to the Kohn–Sham equations are reliant on an iterative calculation to identify the charge density that minimizes the potential energy functional for a given atomic configuration. This process, known as converging the self-consistent field, is the main computational expense within DFT. With a machine learning (ML) model that can effectively bypass the Kohn–Sham equations by accurately and efficiently predicting the charge density, the number of steps required to converge the ground-state electron density can be drastically reduced or potentially eliminated altogether by using the predicted charge density as the initial guess. If accurate enough, a machine-learned charge density could also be used to directly predict electronic structure properties, such as the band gap, band structure, and electronic density of states of a material. Furthermore, the charge density itself can provide an enormous amount of insight into a molecule or material. From the charge density, partial

---

*Correspondence to Xiang Fu (`xiangfu@csail.mit.edu`).

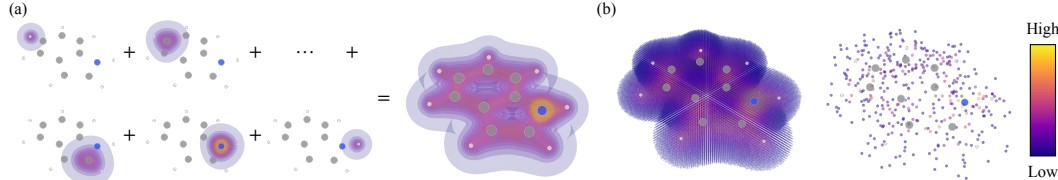

Figure 1: (a) Illustration of the orbital-based method for charge density representation for an example molecule (indole, $C_8H_7N$). The overall charge density is represented as a sum over spherical-harmonics-based atomic orbital basis functions (spherical fields) centered at each atom. (b) Left: Illustration of the probe-based method for charge density representation. The charge density is represented as a voxel where each grid point (probe node) represents a scalar density at that coordinate. The voxel for the example molecule is of size $108 \times 96 \times 40$. Grid points with very small charge densities ($< 0.05$) are not visualized. Right: For a probe-based machine learning prediction model, the voxel contains too many grid points to be processed simultaneously. Sampling of the voxel points is needed during training and inference. All charge densities use the same colormap scale at the right-most side of the figure. Atom color code: H (white), C (gray), N (blue). The charge density is from the QM9 charge density dataset [7].

atomic charges, dipole moments, atomic spin densities, and effective bond orders can all be directly computed through one of several population analysis methods [3, 4]. For some materials discovery tasks, the charge density can also be a crucial descriptor depending on the application area [5, 6]. Therefore, efficient and accurate representations and ML models for charge density prediction are highly desirable as a means of accelerating the discovery of promising molecules and materials.

In machine learning workflows, the charge density is a volumetric, data-rich object, usually represented as voxels with a grid resolution of around 0.1 Å [7, 8]. This poses a challenge, as even relatively small molecules and materials can require hundreds of thousands to millions of grid points to represent the charge density at this (relatively coarse) resolution. At the same time, small deviations in the charge density that result from a representation that is too coarse can have a substantial impact on energy and other derivable properties. This need for both efficiency and accuracy creates a significant challenge for ML methods.

The existing literature has mainly focused on two approaches to learning to predict charge density. The first approach (orbital-based), illustrated in Figure 1 (a), is to predict atomic orbital basis set coefficients by regressing over coefficients extracted from DFT data [9, 10, 11, 12, 13]. The atomic orbital basis functions are based on the composition of radial functions and spherical harmonics. Under this scheme, the charge density is represented as a set of spherical fields centered around each atom. The real space charge density voxel can be constructed by overlaying the spherical fields and evaluating at each grid point. For orbital-based ML models, both the prediction of the basis set coefficients and the evaluation of the spherical fields are relatively scalable, making this approach efficient at inference time. However, this approach can suffer from sub-optimal accuracy due to the limited representation power of the chosen basis set. In particular, it is challenging for the atom-centered atomic orbitals to model complex electronic structures between atoms.

The second approach (probe-based) [14, 7, 15, 16], illustrated in Figure 1 (b), is to predict the charge density by inserting "probe nodes" at all grid coordinates of the charge density voxel and applying graph message passing between the atoms and these probe nodes. Finally, the scalar charge density at each grid coordinate is predicted through node-wise readout over the probe nodes. This approach, while expressive and accurate, is computationally expensive. To see why, recall that the number of grid points in the charge density voxel is usually very large for even a small atomic system. Conducting neural message passing over millions of nodes is both computationally and memory intensive. The large number of nodes usually requires sampling a subset of grid points from the charge density voxel (Figure 1 (b), right) in each training or inference step [7].

This paper aims to address this accuracy–efficiency dilemma with a new recipe for building representations and ML models for charge density prediction. We identify three key ingredients:

1. We represent the charge density using an atomic orbital basis set (spherical fields centered at each atom) to leverage its efficiency and equivariant properties. Beyond orbitals placed

at the atomic coordinates, we further introduce virtual orbitals to improve expressivity. In other words, we also place spherical fields centered at coordinates other than the atomic centers, while ensuring the placement algorithm is SE(3)-equivariant.

2. We use domain-informed and expressive basis sets. In particular, we construct an even-tempered Gaussian basis from an atomic orbital basis set. This allows us to smoothly control the expressivity of the atomic orbitals and enable flexible accuracy–efficiency trade-offs. We make the basis set exponents learnable to further improve expressivity.

3. We use a high-capacity equivariant neural network architecture (eSCN [17]), which enables efficient training and inference with features of high tensor order for a large dataset.

We apply our recipe to the widely used QM9 charge density benchmark [18, 19, 7]. Our method outperforms existing state-of-the-art methods while being around $30\times$ faster. Furthermore, we can flexibly trade off accuracy and efficiency by adjusting the model/basis size; in doing so, we achieve up to $171\times$ efficiency compared to state-of-the-art methods with only a slight degradation in accuracy. This tunability is valuable, as different applications, material classes, and available computing resources may require drastically different levels of accuracy in the charge density prediction. We conduct an ablation study to justify the significance of each proposed ingredient.

## 2 Related Works

**ML methods for charge density prediction.** Orbital-based methods predict coefficients for the orbital basis set functions to recover the target charge density. Past works have explored Gaussian processes [9] and graph neural networks [10, 11, 12, 13] in small molecules, water, and materials systems. [20] used Jacobi-Legendre expansion — a many-body extension of atomic orbitals — for representing and predicting the charge density. These approaches, while efficient, suffer from lower accuracy in benchmarks such as QM9 [18, 19, 7] and the Materials Project [21, 8] charge density datasets. Probe-based methods, on the other hand, predict the charge density by neural message passing between the atoms and probe nodes at all grid points. These methods [14, 7, 15, 16] have shown superior accuracy in both molecules and materials but suffer from poor scalability, as they require neural processing of millions of probe nodes for molecule/material structures of tens of atoms. Recent works also explored a combination of atomic orbitals and probe-based methods [12] or plane-wave basis sets [22]. However, both methods still require neural message passing with a large number of probe nodes, which limits their scalability. In the present work, we combine virtual nodes, even-tempered Gaussian basis, and trainable basis functions to greatly improve the expressivity of orbital basis functions.

**Equivariant neural networks**. Equivariant neural networks [23, 24, 25, 26, 27, 28, 29, 17] use equivariant representations and processing layers that can preserve rotational and translational symmetries that are critical to atomistic modeling tasks. Equivariant models have shown advantages in ML potentials with respect to the accuracy, sample complexity, and molecular dynamics simulation capabilities [30, 31, 32, 33] in addition to charge density prediction tasks [7, 15]. This is because atomic forces and charge densities are indeed SE(3)-equivariant with regard to the input atomic coordinates. In this work, we leverage recent advances in methods for building more expressive and scalable equivariant architectures [17] to improve the accuracy and scalability of charge density prediction.

## 3 Methods

Our recipe for building ML charge density prediction capabilities involves two complementary aspects: the charge density representation and the prediction model.

### 3.1 Charge Density Representation

**Gaussian-type orbitals (GTOs)** are widely used as basis sets for representing electron configurations in quantum chemistry [34]. They are spherical Gaussian functions centered at atomic coordinates. For an atom $i$ at coordinate $\boldsymbol{r}_i$, a GTO basis function with exponent $\alpha$, angular momentum quantum number (also called tensor order or degree) $l$, and magnetic quantum number $m$ is given by the

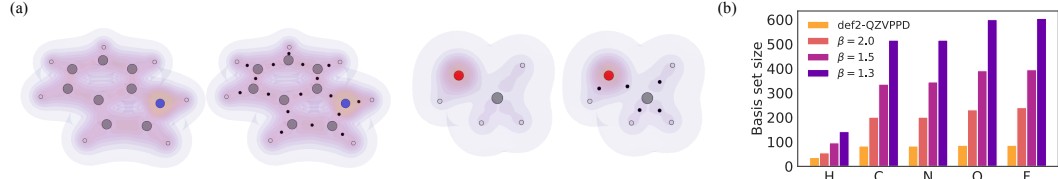

Figure 2: (a) Two example molecules (left: indole, $C_8H_7N$; right: methanol $CH_3OH$), before and after the bond-midpoint-based virtual coordinates (small black points) are inserted. Atom color code: H (white), C (gray), N (blue), O (red), virtual nodes (small, black). (b) The number of Gaussian-type orbital basis functions for selected elements in the `def2-QZVPPD` basis set and even-tempered Gaussian basis sets derived from it under different $\beta$, which controls the number of basis functions as described in Equation (3).

following expression:

$$\Phi_{\alpha,l,m,\boldsymbol{r}_i}(\boldsymbol{r}) \equiv R_l(r)Y_{l,m}\left(\frac{\boldsymbol{r}-\boldsymbol{r}_i}{r}\right) = z_{\alpha,l}\exp(-\alpha r^2)r^l Y_{l,m}\left(\frac{\boldsymbol{r}-\boldsymbol{r}_i}{r}\right), \tag{1}$$

where $r = ||\boldsymbol{r}-\boldsymbol{r}_i||$ is the distance from a query coordinate $\boldsymbol{r}$ to the atom coordinate $\boldsymbol{r}_i$ and $Y_{l,m}$ are real spherical harmonics. $z_{\alpha,l}$ is a normalizing constant, such that $\int_{\mathbb{R}^3}||\Phi||_2^2\,\mathrm{d}V = 1$. For the purpose of developing a machine learning model based on GTOs, we choose to represent the charge density, $\rho$, of an atomic system via a linear combination of many basis functions:

$$\rho(\boldsymbol{r}) = \sum_i^N \sum_j^{N_b^i} \sum_{m=-l_{i,j}}^{l_{i,j}} c_{i,j,m}\Phi_{\alpha_{i,j},l_{i,j},m,\boldsymbol{r}_i}(\boldsymbol{r}), \tag{2}$$

where $N$ is the number of atoms (including virtual ones when applicable), $N_b^i$ is the number of $l$ values ($\alpha$ values) for atom $i$. It should be noted that the charge density in Kohn–Sham DFT is not computed in this way; rather, Equation (2) is an artificial representation for the sake of training a machine learning model that is inspired by the orbital-like character of GTOs. The basis functions $\Phi_{\alpha_{i,j},l_{i,j},m,\boldsymbol{r}_i}$ are chosen first as the basis set, with a fixed set of $l$ and $\alpha$ values for each element. For example, the values for $l$ and $\alpha$ for hydrogen in the `def2-QZVPPD` basis set [35] are presented in Appendix A, Table 2. The number of basis functions for an atom $i$ can be derived as $\sum_{j=1}^{N_b^i}(2\cdot l_{i,j}+1)$, because $m$ can be an integer from $-l$ to $l$. A higher $l$ value corresponds to a more complex angular part of the basis function and allows the corresponding spherical field to be more anisotropic. The number of orbital basis functions for elements H, C, N, O, and F of the `def2-QZVPPD` basis set (and its even-tempered variant, detailed later in this section) is included in Figure 2. Atoms with more complex electronic structures are often represented with more basis functions. The number of basis functions, $l$s, and $\alpha$ values are carefully chosen in existing basis sets such as `def2-QZVPPD`. We refer interested readers to the original papers [35, 36, 37] for more details regarding the construction of atomic orbital basis sets.

In training the machine learning model, after the basis set is determined, the coefficients $c_{i,j,m}$ are then fit such that Equation (2) best represent the charge density. GTOs have been studied in several previous works [9, 11, 12] as a means of representing the charge density with promising results. However, their accuracy still bears significant room for improvement. We next introduce virtual orbitals, even-tempered Gaussian basis, and scaling factors for orbital exponents that greatly improve the expressive power of GTOs for charge density representation.

**Virtual orbitals.** The atom-centered spherical fields often struggle to capture non-local electronic structures, which induces representation errors. This limitation is effectively addressed with the introduction of virtual orbitals, which define sets of spherical fields located in a position other than the atomic centers. Due to the critical importance of chemical bonds in defining the overall electronic structure, we insert virtual nodes into the midpoint of all chemical bonds for a given molecule (illustrated in Figure 2 (a)). With this method, the coordinates to insert the virtual nodes are SE(3)-equivariant with regard to the input atom coordinates. Therefore, as long as the prediction of the basis set coefficients is SE(3)-equivariant, the overall charge density prediction will still be equivariant after the introduction of the virtual orbitals. We discuss potential extensions to virtual

orbital assignments in Section 5. After the virtual nodes are created, one must decide which basis functions to use for the virtual orbitals. In this work, we use the basis functions of element O for the virtual nodes, which offers a balance in accuracy and efficiency based on preliminary experiments.

**Even-tempered Gaussian basis.** The number of basis functions in existing basis sets, such as `def2-QZVPPD`, may be insufficient for representing complex charge densities. At the same time, expanding the number of basis functions requires care in choosing the values of $l$ and $\alpha$ that improve expressivity effectively. As an example, the `def2-QZVPPD` basis set for hydrogen already contains basis functions with $l = 1$ and $\alpha = 2.292$. Extending this basis set with basis functions with $l = 1$ and $\alpha = 2.0$ will not significantly improve its expressivity because the spherical pattern will be similar to existing basis functions. A general methodology for controlling the basis set size is to use an even-tempered Gaussian basis set [38]. Based on a reference atomic orbital basis set (e.g., `def2-QZVPPD`), the even-tempered basis set constructs a series of GTOs with a set of angular momentum quantum numbers $l$ determined by the atomic number and exponents $\alpha$ given by:

$$\alpha_k = \alpha \cdot \beta^k \quad \text{for } k = 0, 1, 2, \ldots, N_l. \tag{3}$$

For each spherical harmonics degree $l$, $\alpha$ and $N_l$ are chosen such that the exponents in the reference atomic orbital basis set are well-covered[2]. $\beta$ controls the number of basis functions — a smaller $\beta$ creates a more expressive basis set with denser exponents. The use of an even-tempered basis set allows us to smoothly control the number of basis functions $N_b^i$ effectively. Figure 2 (b) shows how the number of orbital basis functions for elements H, C, N, O, and F grows with a smaller $\beta$ for the even-tempered Gaussian basis derived from the `def2-QZVPPD` basis set.

**Scaling factors for orbital exponents.** In existing orbital-based models [9, 10, 11, 12, 13], while the coefficients for the basis functions are predicted by the ML model, the exponents are fixed for each atom type and not trainable. However, atoms in different local atomic environments can exhibit significantly different charge density patterns around them, especially for the virtual orbitals that aim at capturing interatomic interactions. To further improve the expressivity of the basis set, we make the exponents trainable by learning a positive scaling factor $s > 0$, such that Equation (1) becomes:

$$\Phi_{\alpha,l,m,\boldsymbol{r}_i}(\boldsymbol{r}, s) = z_{\alpha,l,s} \exp(-s \cdot \alpha r^2) r^l Y_{l,m}\left(\frac{\boldsymbol{r} - \boldsymbol{r}_i}{r}\right). \tag{4}$$

where $z_{\alpha,l,s}$ is a normalizing constant such that $\int_{\mathbb{R}^3} ||\Phi||_2^2 \, dV = 1$. The charge density is now represented with coefficients $c_{i,j,m}$ and scaling factors $s_{i,j}$ as:

$$\rho(\boldsymbol{r}) = \sum_i^N \sum_j^{N_b^i} \sum_{m=-l_{i,j}}^{l_{i,j}} c_{i,j,m} \Phi_{\alpha_{i,j},l_{i,j},m,\boldsymbol{r}_i}(\boldsymbol{r}, s_{i,j}), \tag{5}$$

the introduction of the learnable scaling factors for the exponents significantly improves the expressive power of our charge density representation but is also prone to instability during training. We resolve the instability issue with a fine-tuning approach detailed in Section 3.2.

## 3.2 Prediction Model

Using the atomic orbital basis set representation of charge density, the prediction model aims to predict the basis set coefficients $c_{i,j,m}$ and the scaling factors $s_{i,j}$ for each real and virtual node such that the predicted charge density matches the ground truth density obtained from DFT calculations. The model $F$ takes as input the types $\mathcal{A} = \{\boldsymbol{a}_i \,|\, i = 1, \ldots, N\}$ and coordinates $\mathcal{R} = \{\boldsymbol{r}_i \,|\, i = 1, \ldots, N\}$ of all real and virtual nodes:

$$\{c_{i,j,m}, s_{i,j} | i = 1, \ldots, N; j = 1, \ldots, N_b^i; m = -l_{i,j}, \ldots, l_{i,j}\} = F(\mathcal{A}, \mathcal{R}). \tag{6}$$

**Backbone architecture.** Our construction of the ML prediction model is motivated by the following:

---

[2]We adopt the implementation of PySCF [39] and refer interested readers to the original paper/code for more details on the construction of even-tempered Gaussian basis.

- Charge density is SE(3)-equivariant with regard to the input atom coordinates. An equivariant model that can preserve this symmetry is desired. Concretely, the basis set coefficients $c_{i,j,m}$ are SE(3)-equivariant with regard to the input atom coordinates. The scaling factors $s_{i,j}$ are SE(3)-invariant with regard to the input atom coordinates.

- Charge density is data-rich and very sensitive to the local atomic environment. A high-capacity and expressive model is desired.

- Efficiency is key for general applications of charge density prediction. The model should be efficient while being expressive.

Based on these criteria, we consider equivariant model architectures and the balance between capacity and efficiency. For equivariant models, an important aspect of model expressivity is the representation of node and edge features in the form of irreducible representations (irreps) of SO(3)[3]: spherical harmonic coefficients. A higher degree of representation ($L$) is desired for building high-capacity models. Previous works have employed the PaiNN architecture [27, 7] that is based on Cartesian features (equivalent to $L = 1$) or architectures based on irreps of SO(3) and tensor products [10, 15, 12, 28] for charge density prediction. However, these models suffer from limited expressivity or scalability. Cartesian features are limited in representing angular information ($L = 1$); meanwhile, the $O(L^6)$ complexity of tensor products limits the degree of representation that can be used while remaining computationally feasible.

In this work, we adopt the equivariant spherical channel network (eSCN) architecture [17] as our model backbone. While using SE(3)-equivariant representations and processing layers, the convolution layers in eSCN reduce the SO(3) convolutions [41] or tensor products [23, 31] to convolutions in SO(2) that are mathematically equivalent. It reduces the complexity of the convolution operation from $O(L^6)$ to $O(L^3)$. Further, the use of point-wise, spherical non-linearity in eSCN also distinguishes itself from e3nn-based equivariant models that only apply non-linearity to the scalar features in the irreps. In our experiments, we also find that eSCN outperforms alternative architectures, such as tensor field networks [15, 23] and MACE [42]. Using eSCN as the backbone architecture, we get the last-layer latent features $\boldsymbol{x}_i$ for all real/virtual nodes:

$$\{\boldsymbol{x}_i | i = 1, \dots, N\} = \text{eSCN}(\mathcal{A}, \mathcal{R}). \tag{7}$$

**Prediction layers.** The features $\boldsymbol{x}_i$ are encoded using multi-channel spherical harmonic coefficients (irreps). Note that the prediction target, basis set coefficients $c_{i,j,m}$, are also encoded as multi-channel spherical harmonic coefficients. For example, for an eSCN with $L = 3$ and a latent dimension of 128, the last-layer latent atom features will be `128x0e + 128x1o + 128x2e + 128x3o`. For the (uncontracted) `def2-QZVPPD` basis set of hydrogen described in Table 2, its irreps are `7x0e + 4x1e + 2x2e + 1x3e` (even parity as charge density is reflection-invariant). The scaling factors $s_{i,j}$ are SE(3)-invariant and can be seen as scalar features of multi-channel irreps (`14x0e` for the `def2-QZVPPD` basis set of hydrogen). Therefore, we can make equivariant predictions of the basis set coefficients and invariant predictions of the scaling factors for each atom $i$ through a fully connected tensor product layer over the atom features and additional processing:

$$\{c_{i,j,m}, \boldsymbol{h}_i | j = 1, \dots, N_b^i; m = -l_{i,j}, \dots, l_{i,j}\} = \text{FullyConnectedTensorProduct}(\boldsymbol{x}_i, \boldsymbol{x}_i) \tag{8}$$

$$\{s_{i,j} | j = 1, \dots, N_b^i\} = C_1 / (1 + \exp(-\text{Linear}(\boldsymbol{h}_i) + \ln C_2)) + C_3. \tag{9}$$

The basis set coefficients are directly obtained through the fully connected tensor product. The tensor product also produces scalar features $\boldsymbol{h}_i$ (`128x0e` for a 128-channel eSCN), which are used for predicting the scaling factors. The parameterization of Equation (9) allows the prediction to range from $(C_3, C_1 + C_3)$, and the scaling factors will be $C_1 / (1 + C_2) + C_3$ when the linear network in Equation (9) is zero-initialized. By setting $C_1 = 1.5, C_2 = 2$, and $C_3 = 0.5$, we can limit the range of the scaling factors to be $(0.5, 2)$ (at most halve or double an exponent) and let initial scaling factors be 1 with a zero initialization of the linear layer in Equation (9).

With the predicted coefficients and scaling factors, the charge density prediction $\hat{\rho}$ can be obtained efficiently by evaluating Equation (5) at all grid coordinates of the charge density voxel. We train the model end-to-end with a mean-absolute error loss $\mathcal{L}$ over the charge density:

---

[3]We refer interested readers to [28] and [40] for more information on equivariant geometric neural networks.

Table 1: QM9 charge density prediction error and efficiency on the test set. Metrics for baseline models are from previous papers whenever possible and skipped (-) when unavailable. The metrics ($\downarrow$ means lower the better, $\uparrow$ means higher the better) of the best-performing model are **bold**. The metrics are reported with corresponding standard errors when available. For SCDP models, $K$ is the number of interaction layers in the eSCN backbone, $L$ is the tensor order of the feature representation in the eSCN backbone, and $\beta$ controls the expressiveness of the even-tempered Gaussian basis set. A higher $K$, higher $L$, or lower $\beta$ indicates a more expressive model. eSCN + VO indicates that virtual orbitals are used. NMAE stands for normalized mean absolute error. Efficiency is measured by molecule per minute (mol. per min.).

| | NMAE [%] $\downarrow$ | NMAE, Split 2 [%] $\downarrow$ | Mol. per min. [$\min^{-1}$] $\uparrow$ |
|---|---|---|---|
| i-DeepDFT [7] | $0.357 \pm 0.001$ | - | - |
| e-DeepDFT [7] | $0.284 \pm 0.001$ | - | - |
| ChargE3Net [15] | $0.196 \pm 0.001$ | $0.203 \pm 0.003$ | 3.95 |
| InfGCN [12] | $0.869 \pm 0.002$ | 0.93 | 72.00 |
| InfGCN, GTO only [12] | - | 3.72 | - |
| GPWNO [22] | - | 0.73 | - |
| *SCDP models (Ours)* | | | |
| eSCN, $K = 4, L = 3, \beta = 2.0$ | $0.504 \pm 0.001$ | $0.514 \pm 0.003$ | 675.47 |
| eSCN, $K = 8, L = 6, \beta = 2.0$ | $0.434 \pm 0.006$ | $0.452 \pm 0.017$ | 567.19 |
| eSCN, $K = 8, L = 6, \beta = 1.5$ | $0.381 \pm 0.001$ | $0.391 \pm 0.002$ | 442.25 |
| eSCN + VO, $K = 8, L = 6, \beta = 2.0$ | $0.237 \pm 0.001$ | $0.250 \pm 0.002$ | 231.21 |
| eSCN + VO, $K = 8, L = 6, \beta = 1.5$ | $0.206 \pm 0.001$ | $0.220 \pm 0.002$ | 177.14 |
| eSCN + VO, $K = 8, L = 6, \beta = 1.3$ | $0.196 \pm 0.001$ | $0.209 \pm 0.002$ | 136.92 |
| *SCDP models fine-tuned with scaling factors (Ours)* | | | |
| eSCN, $K = 4, L = 3, \beta = 2.0$ | $0.432 \pm 0.001$ | $0.438 \pm 0.003$ | 644.00 |
| eSCN, $K = 8, L = 6, \beta = 2.0$ | $0.369 \pm 0.007$ | $0.386 \pm 0.018$ | 544.56 |
| eSCN, $K = 8, L = 6, \beta = 1.5$ | $0.346 \pm 0.001$ | $0.354 \pm 0.002$ | 419.57 |
| eSCN + VO, $K = 8, L = 6, \beta = 2.0$ | $0.207 \pm 0.001$ | $0.220 \pm 0.002$ | 221.19 |
| eSCN + VO, $K = 8, L = 6, \beta = 1.5$ | $0.187 \pm 0.001$ | $0.200 \pm 0.002$ | 164.94 |
| eSCN + VO, $K = 8, L = 6, \beta = 1.3$ | $\mathbf{0.178 \pm 0.001}$ | $\mathbf{0.191 \pm 0.002}$ | 125.29 |

$$\mathcal{L} = \mathbb{E}_{\boldsymbol{r} \in \text{Data}} \left[ |\rho(\boldsymbol{r}) - \hat{\rho}(\boldsymbol{r})| \right]. \tag{10}$$

**Fine-tuning for scaling factor prediction.** The scaling factors at the exponents lead to significant training instability when the network is trained from scratch. Therefore, we use a fine-tuning approach, where we first pre-train the model with fixed basis set exponents (an even-tempered Gaussian basis derived from def2-QZVPPD) and then fine-tune the prediction model with a small learning rate with the learning for scaling factors enabled. To achieve this, we zero-initialize the linear layer in Equation (9) and freeze its weights until the fine-tuning stage.

## 4 Experiments

Our experiments on the QM9 charge density benchmark aim to validate the effectiveness of our proposed recipe in both accuracy and efficiency. We refer to our method as **SCDP** models, which stands for **S**calable **C**harge **D**ensity **P**rediction models.

**Dataset and metrics.** The QM9 charge density dataset [18, 19, 7] contains charge density calculations for 133,845 small organic molecules using the Vienna Ab initio Simulation Package (VASP). We adopt the original split, where 123,835, 50, and 10,000 data points are used for training, validation, and testing, respectively. There are on average 18 atoms in each molecule and 666,462 grid points in each charge density voxel. The entire dataset takes 1.1 TB of disk space. Following previous works [7, 15], we benchmark the prediction accuracy with the normalized mean absolute error, defined as:

$$\text{NMAE}(\hat{\rho}) = \frac{\int_{\mathbb{R}^3} |\rho(\boldsymbol{r}) - \hat{\rho}(\boldsymbol{r})| \, \mathrm{d}V}{\int_{\mathbb{R}^3} |\rho(\boldsymbol{r})| \, \mathrm{d}V}, \tag{11}$$

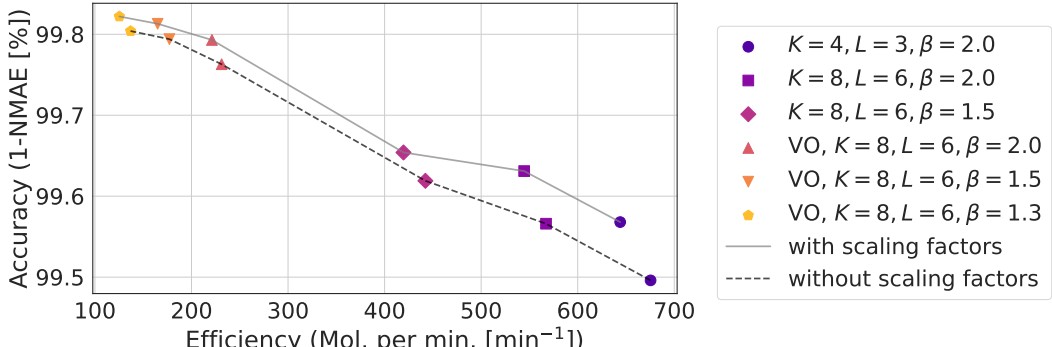

Figure 3: Efficiency–accuracy trade-off for SCDP models. The models with scaling factor fine-tuning form the Pareto front.

where the integration is approximated by summing over the full charge density voxel. We benchmark the efficiency of different methods by the number of molecules predicted per minute (Mol. per min.) on a single NVIDIA A100-80GB-PCIe GPU for the QM9 test split.

In addition to the QM9 charge density dataset, we also benchmark our method on the MD charge density dataset [43, 44, 12] and the Cubic charge density dataset [45]. We use the same data splits as previous works on these benchmarks [12, 22]. Experimental results and comparisons to baselines are included in Appendix A.

**Baseline Models.** We compare SCDP to several previous works on the QM9 charge density prediction benchmark [18, 19, 7]. i-DeepDFT, e-DeepDFT [7], and ChargE3Net [15] are probe-based methods with different backbone architectures: i-DeepDFT uses SchNet [27], e-DeepDFT uses PaiNN [27], while ChargE3Net uses higher-order equivariant features under the tensor field network framework [23, 28]. InfGCN [12] combines GTOs and a shallow network for probe-based inference. It also has a more efficient but less accurate GTO-only variant. GPWNO [22] combines GTOs and plane-wave basis sets but still requires a large number (64,000) of probe nodes for constructing the plane wave prediction. NMAE results for i-DeepDFT, e-DeepDFT, and ChargE3Net are from [15]. NMAE results for InfGC and GPWNO are also from the original papers, which uses a different test split from the default QM9 test split (last 1,600 molecules from the QM9 test split). We benchmark the efficiency of baseline models on our hardware when the source code and pretrained model are publicly available (ChargE3Net and InfGCN). We do not apply any modification to the original code but use optimized configurations for inference to better utilize our GPU: for ChargE3Net, we process 20,000 probes in each batch instead of the default setting of 2,500 probes per batch, and for InfGCN, we process 40,000 probes in each batch with a batch size of 4.

**A significant advance in both accuracy and efficiency.** The metrics for all methods are presented in Table 1. We have a series of SCDP models with different model sizes, basis set sizes, as well as options on the inclusion of virtual orbitals and scaling factors. Our best-performing model uses the virtual orbitals described in Section 3.1, an eSCN of 8 layers and feature representation of order $L = 6$, an even-tempered Gaussian basis with $\beta = 1.3$, and scaling factor fine-tuning. This model achieves an NMAE of 0.178 on the QM9 charge density test set, outperforming the state-of-the-art method ChargE3Net [15] — a probe-based method. While being more accurate, our best model also significantly outperforms ChargE3Net by $31.7\times$ in efficiency. Other configurations of our model with smaller model sizes, basis set sizes, and models without virtual orbitals can trade off accuracy for further gains in efficiency. The trade-off curves are visualized in Figure 3. Compared to a more efficient baseline model, InfGCN, all benchmarked configurations of our method are more efficient and significantly outperform in accuracy. These results convincingly demonstrate a significant advance in the accuracy–efficiency trade-off in ML methods for charge density prediction. Figure 5 in Appendix A shows the convergence of validation NMAE during pretraining and fine-tuning of SCDP models. More details on the hyperparameters for model construction and training are included in Appendix A, Table 3.

**Ablation Analysis.** We discuss the effectiveness of all ingredients through an ablation analysis of the performance of different SCDP models. Starting from the most lightweight model with

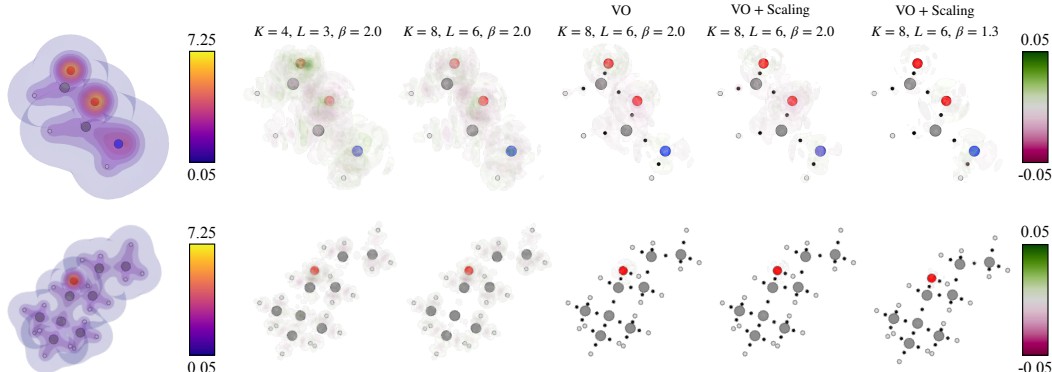

Figure 4: Visualization of the reference charge density and prediction errors for select SCDP models with two representative test molecules (top: $C_2H_3NO_2$ and bottom: $C_8H_{18}O$). The first column is the ground truth charge density with the corresponding color scale. The next five columns are prediction errors from various models which all use the same color scale in the rightmost for error magnitude. The prediction errors significantly reduce with larger model size, virtual orbitals, orbital exponent scaling, and a larger basis set. VO stands for virtual orbitals. Scaling stands for scaling factor fine-tuning. The virtual orbitals significantly reduce errors around chemical bonds. Atom color code: H (white), C (gray), N (blue), O (red), virtual nodes (small, black).

$K = 4, L = 3, \beta = 2.0$ and no virtual orbitals, we first observe that increasing the model size to $K = 8, L = 6$ significantly improves the performance, reducing the NMAE from 0.504% to 0.434%. Next, we increase the basis set size by adjusting $\beta$ from 2.0 to 1.5, which further reduces the NMAE to 0.381%. The introduction of the virtual orbitals renders a significant gain in accuracy by reducing the error from 0.434% to 0.237% for $\beta = 2.0$ and from 0.381% to 0.196% for $\beta = 1.5$. In particular, the charge density near chemical bonds is significantly more accurate after introducing the virtual orbitals, as visualized in Figure 4. On the other hand, for models with higher capacity, the improved accuracy comes at the cost of efficiency. As shown in Table 1 and Figure 3, higher capacity consistently improves performance while sacrificing efficiency. At the same time, all SCDP models remain highly efficient compared to baseline models. When the scaling factors are introduced, accuracy further improves at a slight cost on efficiency for all models. As shown in Figure 3, models with scaling factors from the Pareto front of all SCDP models benchmarked.

## 5   Discussion

Charge density is a fundamental quantity for atomic systems and is central to DFT. ML methods for charge density prediction are promising as a means of greatly accelerating DFT by circumventing the iterative procedure used to find the ground-state charge density given a set of atomic coordinates. In this paper, we propose a recipe that combines three ingredients: (1) virtual nodes; (2) expressive basis sets; and (3) high-capacity equivariant networks that collectively outperform state-of-the-art methods in accuracy while being more than an order of magnitude faster. Nevertheless, there are still many directions for further improving the performance of our proposed model. First, the simple heuristic of assigning virtual node coordinates to bond centers may not be optimal. With recent advances in auto-regressive [46] and diffusion-based [47, 48] equivariant generative models for 3D atomic structures, learning to insert the virtual orbitals may be a promising avenue for optimizing the placement of virtual nodes, thus improving charge density prediction. Due to the "nearsighted" nature of electronic matter [49], an automated method for placing a higher density of virtual nodes near sites of chemical relevance may also be worthwhile to pursue. Second, we can use basis functions beyond Gaussian-type orbitals (e.g., Slater-type orbitals [50, 51] or non-decay radial basis functions [52]) that may require fewer functions to achieve the same level of accuracy.

There are several limitations of the current paper that we aim to address in future work: (1) Despite substantial improvements in efficiency, the computational cost for training the current model is still significant: our best-performing model was pretrained for six days and fine-tuned for six days over four NVIDIA A100 GPUs for the QM9 charge density prediction task. The scaling factor fine-tuning

stage requires a small learning rate, which prolongs training. The prediction model can benefit from resolving the training instability issues with the scaling factors as well as further improvement on the model architecture [53]. (2) While our approach achieves state-of-the-art performance on the QM9 charge density prediction benchmark, its effectiveness in crystalline materials [21, 8] has major room for improvement. The GTOs and the equivariant network can be applied to materials without modification. The bond-midpoint-based virtual node assignment for molecules can be generalized to crystals through a crystal graph construction algorithm, such as CrystalNN [54]. Alternatively, virtual nodes can be iteratively added to occupy void space inside the unit cell of the material using an algorithm based on the Voronoi diagram [55]. The virtual nodes are expected to play an even more important role in prediction accuracy — this is because the diverse atomic species in materials and their complex interactions induce even more complex charge density patterns. (3) To better validate the practical utility of the predicted charge density, evaluation on the reduction of self-consistent field calculations, or on recovering physical observables, such as energy and forces [7, 56, 15], will be highly valuable.

## Acknowledgments and Disclosure of Funding

We thank Teddy Koker, Chaoran Cheng, and Aria Mansouri Tehrani for their helpful discussions and insights. This work was supported by the GIST-MIT Research Collaboration grant funded by GIST and the Machine Learning for Pharmaceutical Discovery and Synthesis (MLPDS) consortium. A.S.R. acknowledges support via a Miller Research Fellowship from the Miller Institute for Basic Research in Science, University of California, Berkeley.

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

# A  Appendix

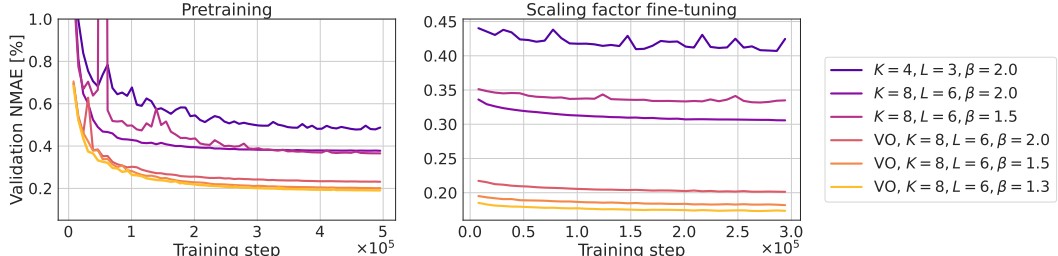

Figure 5: Convergence of validation NMAE during pretraining and finetuning.

Table 2: The (uncontracted) `def2-QZVPPD` basis set for H.

| $l$ | $\alpha$ |
|---|---|
| 0 | 190.6916900 |
| 0 | 28.6055320 |
| 0 | 6.5095943 |
| 0 | 1.8412455 |
| 0 | 0.59853725 |
| 0 | 0.21397624 |
| 0 | 0.080316286 |
| 1 | 2.29200000 |
| 1 | 0.83800000 |
| 1 | 0.29200000 |
| 1 | 0.084063199228 |
| 2 | 2.06200000 |
| 2 | 0.66200000 |
| 3 | 1.39700000 |

---

**Algorithm 1** Pseudo code for the charge density prediction procedure

---

1: **Input**: Atomic numbers $\boldsymbol{A} : (N, 1)$, atom positions $\boldsymbol{R} : (N, 3)$, grid positions $\boldsymbol{R}_c : (M, 3)$
2: **Output**: charge density at the grid positions $\boldsymbol{C} : (M, 1)$
3: obtain atomic irreps features: $\boldsymbol{X} = \{\boldsymbol{x}_i | i = 1, \ldots, N\} = \text{eSCN}(\boldsymbol{A}, \boldsymbol{R})$
4: obtain basis set coefficients $\{c_{i,j,m}, \boldsymbol{h}_i | i = 1, \ldots, N, j = 1, \ldots, N_b^i; m = -l_{i,j}, \ldots, l_{i,j}\}$ and scaling factors $\{s_{i,j} | i = 1, \ldots, N, j = 1, \ldots, N_b^i\} = C_1 / (1 + \exp(-\text{Linear}(\boldsymbol{h}_i) + \ln C_2)) + C_3$ following Equation (8) and (9)
5: **for** $\boldsymbol{r}$ in $\boldsymbol{R}_c$ **do**                    ▷ in practice, we use batched inference
6:     obtain $\rho(\boldsymbol{r})$ following Equation (4) and (5)
7: **end for**
8: **return** $\boldsymbol{C} = \{\rho(\boldsymbol{r}) | \boldsymbol{r} \in \boldsymbol{R}_c\}$

---

**Experimental results on MD and Cubic.** We benchmark the proposed SCDP models on the MD [43, 44, 12] and the Cubic charge density dataset [45] in Table 4. We find the SCDP models significantly outperform baseline models. Hyperparameters used for the MD and Cubic experiments are summarized in Table 5 and Table 6. The MD models are trained on 4 GPUs, while the Cubic model is trained on 8 GPUs. For the molecules in the MD dataset, we use bond centers as coordinates for virtual nodes. For the materials in the Cubic dataset, we iteratively insert virtual nodes up to the number of atoms in the unit cell using an algorithm based on Voronoi diagrams [55].

**Software.** Basis-set-exchange-v0.9.1 [36, 37] and PySCF-v2.5.0 [39] are used to build the orbital basis sets. E3NN-v0.5.1 [28], PyTorch-v1.13.1 [59], and CUDA-v11.6 [60] are used to build the SCDP models. The eSCN [17] implementation is adopted from the Open Catalyst Project [61]. We also acknowledge Numpy [62], ASE [63], Pymatgen [64], wandb [65], Matplotlib [66], and Plotly [67].

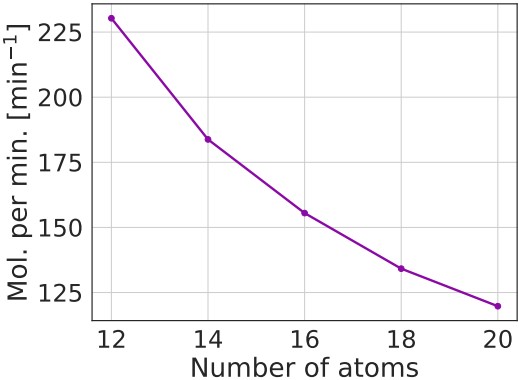

Figure 6: Efficiency as a function of molecular size for our most expressive model (eSCN + VO, $K = 8, L = 6, \beta = 1.3$, with scaling factors). We measure the efficiency by running inference over 500 sampled molecules from the QM9 charge density dataset for a given number of atoms.

## B  Broader Impact

This paper proposes an ML method for accelerating charge density prediction, a crucial task in computational chemistry. The adoption of our method is useful for scientific discovery and can yield positive or negative repercussions, contingent on the applications. The proposed method should be used for materials and drug discovery research that benefit our society.

Table 3: Hyperparameters for SCDP models on the QM9 dataset. [1]The cutoff distance used for building the message passing graph. [2]The cutoff distance for computing the charge density using Equation (5). An orbital basis function only influences all grid coordinates within this distance.

| Hyperparameter | Value |
| --- | --- |
| ***eSCN*** | |
| # interaction layers | $[4, 8]$ |
| $L_{\max}$ | $[3, 6]$ |
| $m_{\max}$ | 2 |
| sphere channels | 128 |
| hidden channels | 256 |
| edge channels | 128 |
| # sphere samples | 128 |
| radius cutoff[1] | 6 Å |
| ***Basis set*** | |
| reference basis set | `def2-QZVPPD` [35] |
| $\beta$ | $[2.0, 1.5, 1.3]$ |
| orbital inference cutoff[2] | 5 Å |
| ***Training*** | |
| batch size | 4 |
| # grid point samples (training, without VO) | $100,000$ |
| # grid point samples (validation/testing, without VO) | $200,000$ |
| # grid point samples (training, with VO) | $60,000$ |
| # grid point samples (validation/testing, with VO) | $120,000$ |
| precision | 32 |
| gradient clipping | 0.5 |
| # training steps (pretraining) | $500,000$ |
| # training steps (fine-tuning) | $300,000$ |
| optimizer | Adam [57] |
| Adam $\beta_1$ | 0.9 |
| Adam $\beta_2$ | 0.999 |
| Adam $\epsilon$ | $1 \times 10^{-8}$ |
| weight decay | 0 |
| initial learning rate (pretraining) | 0.001 |
| initial learning rate (fine-tuning) | $2 \times 10^{-5}$ |
| learning rate scheduler | exponential ($\mathrm{LR} = \text{initial LR} \times 0.96^{\mathrm{step}/C}$) |
| terminal learning rate (pretraining) | $1 \times 10^{-5}$ |
| terminal learning rate (fine-tuning) | $2 \times 10^{-6}$ |
| ***Inference*** | |
| batch size (without VO) | 8 |
| batch size (with VO) | 4 |
| Max # grid points in a forward pass for Equation (5) | |
| eSCN, $K = 4, L = 3, \beta = 2.0$ | $2,000,000$ |
| eSCN, $K = 8, L = 6, \beta = 2.0$ | $1,000,000$ |
| eSCN, $K = 8, L = 6, \beta = 1.5$ | $1,000,000$ |
| eSCN + VO, $K = 8, L = 6, \beta = 2.0$ | $600,000$ |
| eSCN + VO, $K = 8, L = 6, \beta = 1.5$ | $400,000$ |
| eSCN + VO, $K = 8, L = 6, \beta = 1.3$ | $400,000$ |

Table 4: Benchmark results (NMAE) on the MD and Cubic datasets.

| Molecule | SCDP (Ours) | GPWNO [22] | InfGCN [12] |
|---|---|---|---|
| MD-ethanol | $\mathbf{2.34 \pm 0.25}$ | 4.00 | 8.43 |
| MD-benzene | $\mathbf{1.13 \pm 0.06}$ | 2.45 | 5.11 |
| MD-phenol | $\mathbf{1.29 \pm 0.07}$ | 2.68 | 5.51 |
| MD-resorcinol | $\mathbf{1.35 \pm 0.08}$ | 2.73 | 5.95 |
| MD-ethane | $\mathbf{2.05 \pm 0.12}$ | 3.67 | 7.01 |
| MD-malonaldehyde | $\mathbf{2.71 \pm 0.60}$ | 5.32 | 10.34 |
| Cubic | $\mathbf{2.59 \pm 0.25}$ | 7.69 | 8.98 |

Table 5: Hyperparameters for SCDP models on the MD dataset. Parameters that are the same as the QM9 models are omitted in this table.

| Hyperparameter | Value |
|---|---|
| ***eSCN*** | |
| # interaction layers | 4 |
| $L_{\mathrm{max}}$ | 3 |
| $m_{\mathrm{max}}$ | 2 |
| ***Basis set*** | |
| reference basis set | `def2-QZVPPD` [35] |
| $\beta$ | 1.5 |
| orbital inference cutoff[2] | 5 Å |
| ***Training*** | |
| batch size | 4 |
| # grid point samples (training, with VO) | $125,000$ |
| # grid point samples (validation/testing, with VO) | $125,000$ |
| # training steps (pretraining) | $250,000$ |
| # training steps (fine-tuning) | $50,000$ |
| ***Inference*** | |
| batch size (with VO) | 4 |
| Max # grid points in a forward pass for Equation (5) | |
| eSCN, $K = 4, L = 3, \beta = 1.5$ | $1,000,000$ |

Table 6: Hyperparameters for SCDP models on the Cubic dataset. Parameters that are the same as the QM9 models are omitted in this table.

| Hyperparameter | Value |
|---|---|
| ***eSCN*** | |
| # interaction layers | 8 |
| $L_{\max}$ | 4 |
| $m_{\max}$ | 2 |
| ***Basis set*** | |
| reference basis set | def2-universal-JKFIT [58] |
| $\beta$ | 1.5 |
| orbital inference cutoff[2] | 4 Å |
| ***Training*** | |
| batch size | 2 |
| # grid point samples (training, with VO) | $25,000$ |
| # grid point samples (validation/testing, with VO) | $35,000$ |
| # training steps (pretraining) | $500,000$ |
| # training steps (fine-tuning) | 0 |
| ***Inference*** | |
| batch size (with VO) | 2 |
| Max # grid points in a forward pass for Equation (5) | |
| eSCN, $K = 8, L = 4, \beta = 1.5$ | $100,000$ |

Table 7: Hyperparameters for baseline model architectures ($\beta = 2.0$, all other hyperparameters are kept the same as in Table 3).

| Hyperparameter | Value |
|---|---|
| ***Charge3Net backbone*** | |
| # interaction layers | 4 |
| $L_{\max}$ | 3 |
| Feature irreps | 167x0o + 167x0e + 56x1o + 56x1e + 33x2o + 33x2e |
| ***MACE backbone*** | |
| # interaction layers | 4 |
| $L_{\max}$ | 3 |
| Hidden irreps | 64x0e + 64x1o + 64x2e |
| MLP irreps | 128x0e |
| ***Common*** | |
| Max # grid points in a forward pass | $2,000,000$ |
| Initial learning rate | $1 \times 10^{-2}$ |
| Terminal learning rate | $1 \times 10^{-4}$ |

