# OpenReview forum: "A Recipe for Charge Density Prediction"
_NeurIPS.cc/2024/Conference — NeurIPS 2024 poster_

### Official Review · Reviewer_nCiL · 2024-07-07

**Soundness:** 3
**Presentation:** 3
**Contribution:** 3
**Rating:** 8
**Confidence:** 4

**Summary:**

The paper proposes a new method for calculating charge density using orbital based representations along with an equivariant machine learning architecture. The paper starts by introducing density functional theory (DFT) and how charge density plays a central role for DFT. The important role of charge density in DFT motivates the need for a fast, accurate and robust method of calculating it creating an opportunity to apply machine learning to the problem. The introduction also describes how charge density is a volumetric object, which can it difficult to apply ML methods due to data density and how prior methods have aimed to address the overall challenge. After outlining the primary contributions of the paper, Section 2 provides greater detail for related work in ML methods for charge density prediction and equivariant neural networks.

Section 3 describes the primary method, which focuses on charge density representation and the prediction model. The charge density representation includes Gaussian-type orbitals approximated as a linear combination of spherical harmonics basis functions, as well as virtual orbitals to capture non-local electronic structures and interactions. The charge density approximation also includes trainable orbital exponents for each atom type making the representation more expressive. For the prediction model, the paper motivates the need for an equivariant and expressive model architecture and describes the eSCN-based architecture used for the experiments. The last paragraph of the section outlines a pretraining based approach to enhance training stability for the charge density prediction model.

Section 4 describes the experimental results performed on the charge density dataset for the QM9 database of molecules. The results include a comparison to other methods, as well as an ablation analysis related to the different SCPD models' trade-offs for accuracy and computational efficiency. The paper subsequently ends with a discussion along with a description of limitations that can be used for future research work.

**Strengths:**

The paper proposes a new method for a relevant problem in AI for scientific applications. Its strengths include:
* A new method for calculating charge density and integrating ML methods into computational chemistry workflows. [Originality, Significance]
* A detailed description of the method, relevant background and experimental details. [Quality, Clarity]
* Experimental results show convincing improvement compared to the state-of-the art. [Quality, Significance]

**Weaknesses:**

The paper could be improved by providing more detail along the following:
* Provide more details on how charge density prediction is different from ML potentials and why that distinction matters. Concretely, what advantages and disadvantages does charge density prediction have over ML potentials and vice versa? [Significance, Clarity]
* Provide greater clarity on how their proposed method for Charge Density Representation differs from prior approaches, such as a summary table. [Clarity, Significance]
* Address the questions below.

-----------

Assessment updated during discussion period.

**Questions:**

* Were there other options you considered for the def2-QZVPPD basis set? Can you describe your considerations and/or if it's reasonable to perform an ablation related to this? You somewhat discuss this in "Even-tempered Gaussian basis." but its unclear to me if an empirical evaluation is feasible.
* Can you include traditional DFT throughput numbers in the table for reference? This is to get a sense of the speedup.
* Can you describe in more detail why an equivariant network is required? Have you tried evaluating non-equivariant networks?

**Limitations:**

The paper includes a discussion on limitations in Section 5.

---

> ### Author Rebuttal · Authors · 2024-08-06
>
> We thank reviewer nCiL for helpful feedback and comments. We address each of the reviewer’s concerns below.
>
> > Provide more details on how charge density prediction is different from ML potentials and why that distinction matters. Concretely, what advantages and disadvantages does charge density prediction have over ML potentials and vice versa? [Significance, Clarity]
>
> Thank you for the constructive feedback. This is a great point to include in the paper. We briefly summarize several key differences here:
>
> - **Charge density is the core of DFT from which all properties can be derived.** There are many important molecular/materials properties, such as the electronic band structures, dipole moments, atomic spin densities, and effective bond orders, that can be directly computed from the charge density but not from an ML potential. An ML potential can be seen as a coarse-graining of the charge density prediction task where only energy/forces are considered. The charge density represents all the information from DFT calculations.
> - **Self-consistency-field (SCF) from predicted charge density guarantees DFT accuracy.** ML-predicted charge density can be used as an initialization for the SCF calculation of DFT. This can reduce the computational cost of DFT and correctness is guaranteed. An ML potential does not have any performance guarantee.
> - **Detailed charge information matters.** Modeling of charge is still an active research topic in building better ML potentials, especially for reactive systems. Charge density encapsulates this crucial information for high-fidelity modeling.
> - **Charge density Pros**: the finest-grained information from DFT; all chemical properties can be derived; can be combined with conventional DFT for accuracy guarantee; can model complex systems more accurately.
> - **Charge density Cons**: more costly than ML potentials; not required for many relaxation/simulation tasks that don’t require very high accuracy.
>
> We will include these discussions in the final version.
>
> > Provide greater clarity on how their proposed method for Charge Density Representation differs from prior approaches, such as a summary table. [Clarity, Significance]
>
> Thank you for the great suggestion. Our method builds on top of atomic orbital representations of charge density and introduces even-tempered basis, virtual nodes/orbitals, and learnable scaling factors to improve its expressivity while preserving its efficiency. These elements are also highly tunable to trade-off accuracy and efficiency.
>
> We summarize a comparison of the different representations below:
>
> | Method | Expressive? | Efficient? | Flexible trade-off? |
> |---------------------------|-------------|------------|----------------------|
> | Orbital-based (e.g., [1]) | No | Yes | No |
> | Probe-based (e.g., [2]) | Yes | No | No |
>  Ours | Yes | Yes | Yes |
>
>
> We will include these further details in the final version.
>
> > Were there other options you considered for the def2-QZVPPD basis set? Can you describe your considerations and/or if it's reasonable to perform an ablation related to this?
>
> In early small-scale experiments, we have considered other basis sets such as `def2-universal-JKFIT` and `aug-cc-pVQZ`. We choose def2-QZVPPD because it’s one of the most expressive and popular basis sets and its promising initial results. We are unable to complete this experiment at the time of rebuttal due to limitations on time and compute but would be happy to include an ablation study in the final version.
>
> > Can you include traditional DFT throughput numbers in the table for reference?
>
> Unfortunately, the DFT runtime was not included in the QM9 charge density dataset [3]. The throughput of conventional DFT depends a lot on the level of theory, convergent criterion, and the size of the system as it scales as $O(N^3)$. Practical calculations of a small molecule usually take a few minutes to complete, and the run time grows rapidly when the number of atoms increases. ML-based charge density prediction methods scale linearly with the number of atoms, and therefore are promising to enable larger-scale calculations at higher levels of theories.
>
> > Can you describe in more detail why an equivariant network is required? Have you tried evaluating non-equivariant networks?
>
> An equivariant model is highly desired because the charge density is an equivariant quantity. That is, when the molecule is rotated/translated, the charge density should rotate/translate accordingly. When predicting the charge density through the orbital basis representation, the basis set coefficients correspond to different components of spherical harmonics. An equivariant model is able to predict the charge density equivariantly through equivariant prediction of the basis set coefficient. We only evaluated equivariant networks because a non-equivariant network breaks the physical constraints of the charge density prediction task. Different from an ML potential where the energy is an invariant quantity and the forces can be obtained through taking its derivative with regard to atomic positions, the basis set coefficients that we predict are not scalars nor derivatives of scalars, and can be of higher tensor order. Equivariant models are highly suitable for predicting these quantities.
>
> We look forward to further discussions if you have additional questions or suggestions.
>
> [1] Rackers, Joshua A., et al. "A recipe for cracking the quantum scaling limit with machine learned electron densities." Machine Learning: Science and Technology 4.1 (2023): 015027.
>
> [2] Koker, Teddy, et al. "Higher-order equivariant neural networks for charge density prediction in materials." npj Computational Materials 10.1 (2024): 161.
>
> [3] Jørgensen, Peter Bjørn, and Arghya Bhowmik. "Equivariant graph neural networks for fast electron density estimation of molecules, liquids, and solids." npj Computational Materials 8.1 (2022): 183.

---

> > ### Comment · Reviewer_nCiL · 2024-08-10
> >
> > Thank you for the additional details. I think they make the paper stronger and have adjusted my score accordingly. I have one additional suggestions:
> >
> > * For future work or related work, it could be helpful to understand what would be required to cover a greater extend of relevant charge density prediction cases. Are more datasets needed beyond QM9 and MP? If so, what should researchers focus on? What types of other open problems exist that might be interesting for the machine learning community?

---

> ### Author Response · Authors · 2024-08-11
> **Thank you for your response**
>
> Thank you for your response and additional suggestions!
>
> > Are more datasets needed beyond QM9 and MP? If so, what should researchers focus on? What types of other open problems exist that might be interesting for the machine learning community?
>
> These are great questions. More datasets are certainly desirable -- first, the QM9 and MP datasets are not calculated with a very high level of theory -- which is where ML is going to help the most. An interesting future direction would be learning the difference between a higher and a lower level of theory, so the lower level of theory can be used as a prior, which may make the learning task easier so it requires less data/compute to train. It would also be interesting to consider other classes of systems -- QM9 is for small organic molecules, MP is for inorganic crystals -- other systems such as catalytic surfaces and metal organic frameworks could also be considered.
>
> From the few directions proposed in our previous rebuttal response, we believe in mid to short term we will be able to create efficient and accurate ML-based charge density prediction capabilities that can accelerate DFT through accurate SCF initialization, and obtain a variety of properties such as partial charges and dipole moments. We are also eager to see future research in pretraining/co-training of ML potentials or property predictors with charge density data. Addressing the limitations discussed in the paper to further improve charge density prediction is also exciting to us. In long term, we believe charge density modeling is crucial for high-fidelity modeling of more complex scenarios such as excited-state DFT and chemical reaction, and potential impact in designing better DFT functionals.
>
> We truly appreciate your time, consideration, and feedback that greatly help us improve our paper.

---

### Official Review · Reviewer_SS9K · 2024-07-09

**Soundness:** 3
**Presentation:** 3
**Contribution:** 3
**Rating:** 6
**Confidence:** 3

**Summary:**

This paper proposes a new effective approach to estimating the charge density of molecular systems using machine learning models. Although this topic has been recently actively studied, the present approaches suffer from either a lack of accuracy or scalability. The proposed approach alleviated the problem by identifying three key ingredients: (1) utilizing atomic and virtual orbitals, (2) utilizing expressive and learnable orbital basis sets, and (3) utilizing high-capacity equivariant neural network architecture. The proposed approach achieved state-of-the-art accuracy on QM9 while reducing computational cost.

**Strengths:**

1. By combining domain domain-knowledge (GTOs, Even-tempered Gaussian basis, etc) and new machine learning techniques (virtual orbitals, scaling factors for orbital exponents), the method achieves around 10% error reduction with around 30 times better efficacy in comparison to the previous state-of-the-art model (ChargE3Net).

2. "Methods" are well-described with detailed explanations from physical and engineering point of view.

**Weaknesses:**

The following (at least major) weaknesses should be approached before acceptance:

Major weakness (the upper, the higher priority):

1. The paper only performed the experiments on QM9 and I wonder if the method is over-fitted to the QM9 dataset. The previous methods (InfGCN, ChargE3Net, GPWNO) performed their experiments on average 3-dataset, e.g., among NMC, Cubic, MD, MP, and QM9. Since this method's scope is now limited to molecular systems, either MD or a molecule dataset used in [1] can be utilized. Because of the time limitation of the rebuttal period, at least logical validation of the effectiveness of the other dataset would be necessary (Of course the additional experiments are highly recommended, at least before publication).

2. Although not so crucial, the code is not provided in supplemental material. So, at this stage of this review, I do not have confidence that the paper's result can be reproducible. And I wonder why the authors did not submit it even though mentioning they would provide it after acceptance. Are there any proper reasons to hide the code from the reviewers?

3. Even though the author mentions at line 214-215 that eSCN outperforms tensor field networks and MACE, I cannot find the corresponding part in Section 4. An explicit indication of the experiment would be desirable.

[1] Jørgensen, P.B., Bhowmik, A. Equivariant graph neural networks for fast  electron density estimation of molecules, liquids, and solids. npj Comput Mater 8, 183 (2022). https://doi.org/10.1038/s41524-022-00863-y

Minor (for further improvement):

1. The explanation in "Prediction layers" can be improved. It would be strongly recommended to provide more information in the Appendix using more space.

2. From the description of "Prediction layers", this method is a little difficult to apply to other models than eSCN. If not, it would be helpful for the readers that the authors provide additional "recipe" for the other models.

3. The inference time scaling in terms of the molecular size (atom number) is missing, which would be important to apply for larger molecular systems, for example, protein, on which DFT has difficulty performing the calculation.

**Questions:**

1. The idea of the scaling factors for orbital exponents in Equation (4) seems to have a relation with the D4FT [2] Equations (20) and (21). Is the idea independently developed? It would be better to cite the paper and explain the difference.

2. At line 102, roto-translational --> rotation-translational     (the former seems not strictly accepted term by all the community).

3. In Table 1, the "Mol. per min." shows around 10% increases in SCDP when adding scaling factors. Why? Equation (4) indicates that the scaling factor would not increase the model complexity and size so much.

[2]Li, Tianbo, et al. "D4FT: A deep learning approach to Kohn-Sham density functional theory." arXiv preprint arXiv:2303.00399 (2023).

**Limitations:**

Yes. It is provided in the "Discussion" section.

---

> ### Author Rebuttal · Authors · 2024-08-06
>
> We thank reviewer SS9K for helpful feedback and comments. We address each of the reviewer’s concerns below.
>
> > The paper only performed the experiments on QM9...
>
> Thank you for your feedback. We follow your suggestions and extend our experiments to the MD dataset used in [1]. Due to limited time and compute resources, we were only able to complete experiments for four out of six molecules in the MD dataset. We report the NMAPE performance below (baseline metrics adopted from [1]):
>
> | Molecule   | SCDP (Ours) | InfGCN | CNN  | DeepDFT | DeepDFT2 | EGNN  | DimeNet | DimeNet++ | GNO   | FNO   | LNO   |
> |-----|------|--------|------|---------|----------|-------|---------|-----|-------|-------|-------|
> | ethanol    | **2.40 ± 0.26**       | 8.43   | 13.97| 7.34    | 8.83     | 13.90 | 13.99   | 14.24     | 82.35 | 31.98 | 43.17 |
> | benzene    | **1.15 ± 0.06**        | 5.11   | 11.98| 6.61    | 5.49     | 13.49 | 14.48   | 14.34     | 82.46 | 20.05 | 38.82 |
> | phenol     | **1.32 ± 0.07**        | 5.51   | 11.52| 9.09    | 7.00     | 13.59 | 12.93   | 12.99     | 66.69 | 42.98 | 60.70 |
> | resorcinol | **1.38 ± 0.08**        | 5.95   | 11.07| 8.18    | 6.95     | 12.61 | 12.04   | 12.01     | 58.75 | 26.06 | 35.07 |
>
> Our method significantly outperforms baseline models. In all experiments, we use an SCDP model with an eSCN of 4 layers, $L_{\mathrm{max}}=3$, and $\beta=1.5$. We train for 250000 steps while keeping all other training hyperparameters unchanged. A smaller model and shorter training schedule are used because the MD datasets are smaller in size. We will update results on all molecules in the MD dataset once training is completed. We appreciate your understanding of the time limitation of the rebuttal period.
>
> > Although not so crucial, the code is not provided in supplemental material.
>
> We planned to do further improvement on the clarity of our codebase at the time of submission and we are happy to submit our code for peer review. Following NeurIPS 2024 instructions, we have sent an official comment to the AC that includes an anonymous link to the code base.
>
> > Even though the author mentions at line 214-215 that eSCN outperforms tensor field networks and MACE…explicit indication of the experiment would be desirable.
>
> Thank you for the constructive feedback. We find that eSCN performs better initially in small-scale experiments. We provide the performance metrics for a tensor field network backbone used in Charge3Net and MACE, with no virtual nodes and $\beta = 2.0$. We compare these models to the small eSCN model with no virtual nodes and $\beta = 2.0$, so the basis set expressive power is the same. The hyperparameters for the baseline models are adopted from previous works and are included in the rebuttal PDF.
>
> | Model | NMAPE | Mol. per Min. |
> |-|-|-|
> | Charge3Net | 4.79 ± 0.026 | 505.10 |
> | MACE | 5.14 ± 0.024 | 660.25|
> | eSCN | 0.504 ± 0.001 | 675.47 |
>
> We conclude the performance of the eSCN model is significantly better. The source code for the Charge3Net and MACE backbones are also included in our code submission.
>
> > The explanation in "Prediction layers" can be improved.
>
> Thank you for your suggestion. We include pseudo code for the prediction process in the rebuttal PDF and would love to hear it if you have further feedback.
>
> > From the description of "Prediction layers", this method is a little difficult to apply to other models than eSCN.
>
> Our method can be directly applied to any geometric deep learning backbone models that use irreps (irreducible representations of SO(3)) for feature representations, which is common in modern equivariant architectures. To predict the GTO coefficients in an equivariant way, the model needs to output a set of Irreps, that can be obtained through a tensor product of the atom feature irreps.
>
> >The inference time scaling in terms of the molecular size (atom number) is missing.
>
> Thank you for the constructive feedback. In theory, the run time scales linearly ($O(N)$) with regard to the number of atoms (as opposed to an $O(N^3)$ scaling for DFT). We add additional experiments on the inference time scaling in terms of the molecular size using our most accurate model. The results are shown below:
>
> | Number of Atoms | Mol. per Min. |
> |---|---|
> | 12 | 230.31 |
> | 14 | 183.79 |
> | 16 | 155.50 |
> | 18 | 134.17 |
> | 20 | 119.76 |
>
> We also include a plot in the rebuttal PDF. The result shows the runtime indeed scales linearly with regard to the number of atoms.
>
> > The idea of the scaling factors...a relation with the D4FT [2] Equations (20) and (21).
>
> Thank you for pointing out this related work that we were unaware of. Both our method and D4FT aim to enhance the Gaussian basis set through learnable scaling factors – D4FT uses it in representing the wave function for KS-DFT; we use it in representing the charge density. We will update the paper/reference in the final version.
>
> > At line 102, roto-translational --> rotation-translational
>
> Thank you for pointing this out, we will modify it in the final version.
>
> > In Table 1, the "Mol. per min." shows around 10% increases in SCDP when adding scaling factors. Why?
>
> This is a great question. There are two sources of additional compute when the scaling factors are introduced: (1) the network has an extra layer for predicting the scaling factors; (2) In equation (4), when the scaling factors are introduced, the normalization $z_{\alpha, l, s}$ depends on the scaling factor $s$, which now changes in every inference steps. Therefore, the normalization needs to be recomputed in every forward pass with non-negligible computational cost. Without the scaling factors, the normalization only needs to be computed once when initializing the GTOs. This is reflected in L112-L115 of `gtos.py` of our source code.
>
> We look forward to further discussions if you have additional questions or suggestions.
>
> [1] Cheng, Chaoran, and Jian Peng. "Equivariant neural operator learning with graphon convolution." NeurIPS 2023.

---

> > ### Comment · Reviewer_SS9K · 2024-08-07
> > **reply**
> >
> > I really appreciate the authors' effort for the rebuttal. I am basically satisfied with the author's responses.
> >
> > Concerning W1, it is helpful if the authors can tell me which dataset the authors are going to try further for the camera-ready version, in addition to QM9 and MD. It would help me to decide the reconsidered evaluation score.

---

> ### Author Response · Authors · 2024-08-08
> **Thank you for your response**
>
> Thank you for your response! We have completed the experiments for the two other molecules (ethane and malonaldehyde):
>
> | Molecule      | SCDP (Ours) | GPWNO | InfGCN | CNN  | DeepDFT | DeepDFT2 | EGNN  | DimeNet | DimeNet++ | GNO   | FNO   | LNO   |
> |---------------|-------------|--------|---|------|---------|----------|-------|---------|-----------|-------|-------|-------|
> | ethanol       | **2.40 ± 0.26**         | 4.00 | 8.43   | 13.97| 7.34    | 8.83     | 13.90 | 13.99   | 14.24     | 82.35 | 31.98 | 43.17 |
> | benzene       | **1.15 ± 0.06**       | 2.45 | 5.11   | 11.98| 6.61    | 5.49     | 13.49 | 14.48   | 14.34     | 82.46 | 20.05 | 38.82 |
> | phenol        | **1.32 ± 0.07**        | 2.68 | 5.51   | 11.52| 9.09    | 7.00     | 13.59 | 12.93   | 12.99     | 66.69 | 42.98 | 60.70 |
> | resorcinol    | **1.38 ± 0.08**        | 2.73 | 5.95   | 11.07| 8.18    | 6.95     | 12.61 | 12.04   | 12.01     | 58.75 | 26.06 | 35.07 |
> | ethane        | **2.10 ± 0.13**        | 3.67 | 7.01   | 14.72| 8.31    | 6.36     | 15.17 | 13.11   | 12.95     | 71.12 | 26.31 | 77.14 |
> | malonaldehyde | **2.77 ± 0.63**      | 5.32 | 10.34  | 18.52| 9.31    | 10.68    | 12.37 | 18.71   | 16.79     | 84.52 | 34.58 | 47.22 |
>
> We find our method ( 4-layer eSCN, $L_{\mathrm{max}} = 3$, $\beta = 1.5$, with virtual nodes) significantly outperforms all baseline models on all six molecules in the MD dataset. Next, we plan to experiment with scaling factor finetuning for the MD molecules. We are also happy to further extend our experiments to a materials dataset such as NMC or Cubic in the final version.
>
> Thank you again for your time and considerations. We are happy to discuss further if you have additional feedback.

---

> > ### Comment · Reviewer_SS9K · 2024-08-08
> > **reply**
> >
> > Thank you for your fruitful information.
> > I look forward to the camera-ready version and re-evaluate my score.

---

### Official Review · Reviewer_L6ft · 2024-07-11

**Soundness:** 3
**Presentation:** 3
**Contribution:** 2
**Rating:** 4
**Confidence:** 4

**Summary:**

Overall this is a nice technical contribution towards the goal of machine-learning based prediction of charge densities trained from DFT calculations. The main merits are that the authors combine the recent eSCN equivariant network, which reduces the computational complexity of equivariant message passing compared to SO(3) message passing, and other existing ideas such as virtual atom/sites, and learnable (through the exponent rescaling) GTOs. The paper is well written and easy to follow. I have no doubt this is a valuable contribution. But my concern is lack of original ideas (all ideas were existing or straightforward) and lack of high impact application. To me you need to have at least one for acceptance.

**Strengths:**

Motivations are clear. Technical details are well presented and easy to follow. The experiments are also clear.

**Weaknesses:**

Main problem is that the ideas are mainly straight application of existing algorithms and perhaps codes. It is not trivial to put all these pieces together but still I do not see much novelty in ML. The main advantage in application is efficiency compared to e.g. chargeE3Net as the accuracy is very similar. First, the issue of efficiency matters most when one has a killer application for this method. There can be but it is not demonstrated or even explained clearly. ML AIMD might be but we already have ML potentials for that. Second, the efficiency test of ChargeE3Net needs to be explained more clearly. Are we comparing both methods with the same batch size, molecular size, hyperparameters, etc?

**Questions:**

Page 5. "where z_alph... =1" should be in the preceding sentence.
Page 6, equation 9. Are the hidden scalar features h_i the direct scalar output of eSCN? Or the output of FCTensorProd.?
Page 6, lines 230. If I am not mistaken, should be "c3, c1+c3" and "c1/(1+c2)+c3"

**Limitations:**

Yes.

---

> ### Author Rebuttal · Authors · 2024-08-06
>
> We thank reviewer L6ft for helpful feedback and comments. We address each of the reviewer’s concerns below.
>
> > I have no doubt this is a valuable contribution. But my concern is lack of original ideas (all ideas were existing or straightforward) and lack of high impact application.
>
> Thank you for your appreciation of our work as a valuable contribution. We believe our paper makes several significant technical contributions from an ML perspective which enable a significant improvement in model performance. We also believe charge density prediction has many impactful applications. We further elaborate on these two points below.
>
> > Main problem is that the ideas are mainly straight application of existing algorithms and perhaps codes. It is not trivial to put all these pieces together but still I do not see much novelty in ML.
>
> In the context of charge density prediction, the trade-off between accuracy and efficiency is a long-standing problem. We are driven to resolve this challenge and combine several well-motivated novel ideas. Our contributions include many design choices on how to build the GTO basis sets, the use of virtual nodes for charge density representations, expressive prediction networks, and an end-to-end training/finetuning procedure. These methods required a significant amount of trial and error to set up correctly and combined together.
>
> These novel technical contributions lead to a final method that is simple yet very effective, and we see the simplicity of our final model as an advantage. Our experiments also reveal novel insights into understanding the bottleneck of ML-based charge density prediction, as well as model considerations when modeling a data-rich modality. Further, we believe the accuracy, efficiency and flexibility make our proposed method a good stepping stone for future research. We are happy to discuss further if your concerns remain.
>
> > First, the issue of efficiency matters most when one has a killer application for this method. There can be but it is not demonstrated or even explained clearly. ML AIMD might be but we already have ML potentials for that.
>
> In this work, we focus on striking a favorable accuracy-efficiency trade-off as it requires significant domain knowledge and efforts for downstream applications that are independent of the ML contributions. Here we elaborate on the applications of ML-based charge density prediction, and why we believe an accurate+efficient model is particularly impactful.
>
> - **Charge density is the core of DFT from which all properties can be derived.** There are many important molecular/materials properties, such as the electronic band structures, dipole moments, atomic spin densities, and effective bond orders, that can be directly computed from the charge density but not from an ML potential. An ML potential can be seen as a coarse-graining of the charge density prediction task where only energy/forces are considered.
> - **Self-consistency-field (SCF) from predicted charge density guarantees DFT accuracy.** ML-predicted charge density can be used as an initialization for the SCF calculation of DFT. This can reduce the computational cost of DFT and correctness is guaranteed. An ML potential does not have any performance guarantee.
> - **Charge density as pretraining.** Charge density is a data-rich modality that contains all the information from the quantum mechanical calculations, while energy/forces are a small portion of this information. We believe pre-training on charge density is a promising future direction to improve ML potentials, property prediction, and other atomistic modeling tasks.
> - **Multi-modality in atomistic modeling.** Modeling of charge is still an active research topic in building better ML potentials, especially for reactive systems. Modeling the charge density is a promising direction for incorporating charge information into ML potentials. Efficiency is especially important when applying the ML potential in heavy workflows such as molecular dynamics simulations.
> - **An efficient+accurate model unlocks a major roadblock in learning with charge densities.** While the several directions stated above are potentially very impactful, a major roadblock for leveraging the charge density is the lack of a method that is both efficient and accurate. Our contribution addresses this challenge and can potentially make charge density a much more accessible modality for future research.
>
> > The efficiency test of ChargeE3Net needs to be explained more clearly. Are we comparing both methods with the same batch size, molecular size, hyperparameters, etc?
>
> Our method and ChargE3Net are fundamentally different in how we represent charge density, so the batch size is not directly comparable. We report all our hyperparameters in the appendix. Because ChargE3Net is a probe-based model, efficiency is limited by the fact that it needs to conduct neural message passing between all atoms and all positions of the charge density grid, which is very expensive as there are usually hundreds of thousands of grid positions for QM9 molecules.
>
> When comparing efficiency, we apply all models to the same set of molecules (test set of the QM9 charge density dataset). We use the pretrained QM9 model released by the ChargE3Net authors (therefore the original hyperparameters), and use optimized inference parameters to maximize the utilization of our GPU. For ChargE3Net, we process 20,000 probes in each batch which is ~1.3x faster than the default setting of 2,500 probes per batch from the ChargE3Net authors. All model efficiencies are benchmarked on the same machine and GPU.
>
>
> > Typos/clarifications on page 5 and 6
>
> Thank you for pointing out the typos on line179 and 230. We will correct it in the final version. The hidden scalar features $h_i$ is the output of the final FCTensorProduct layer. We will clarify this in the final version.
>
> We look forward to further discussions if you have additional questions or suggestions.

---

> > ### Comment · Reviewer_L6ft · 2024-08-10
> >
> > I appreciate the authors' response. I fully agree with the authors that the ability to predict charge density accurately opens the door to many downstream applications, and never doubted that. My main concern was and still is whether this specific contribution can be demonstrated to be generalizable and useful in applications. I'll keep my rating. Solid technical paper nevertheless.

---

### Official Review · Reviewer_paLd · 2024-07-12

**Soundness:** 4
**Presentation:** 4
**Contribution:** 3
**Rating:** 8
**Confidence:** 3

**Summary:**

This application-oriented paper uses equivariant GNNs to predict charge density, represented orbital basis sets.

**Strengths:**

- the idea of this paper is sound. Representing the charge density using atomic orbital basis set sounds more efficient than the voxel-based methods.
- the performance of this new method is evidenced by well-designed experiments
- the presentation of this paper is excellent

**Weaknesses:**

- I feel like the architectures are a simple variant of popular backbones. But this shouldn't affect the novelty of this application paper.
- the computation cost is still considerable

**Questions:**

You mentioned in the text that if you use a different backbone other than eSCN, the performance is worse. It would be great if you could share some numbers to back up this statement.

**Limitations:**

The limitations are sufficiently discussed in the paper.

---

> ### Author Rebuttal · Authors · 2024-08-06
>
> We thank reviewer paLd for helpful feedback and comments. We address each of the reviewer’s concerns below.
>
> > I feel like the architectures are a simple variant of popular backbones. But this shouldn't affect the novelty of this application paper.
>
> We agree the model backbone architecture is a simple variant of the popular eSCN backbone. We believe the novelty of this paper lies in the combination of several representation, prediction, and training techniques – and most importantly, the significant improvements in model performance, which unlocks many directions for future research.
>
> > the computation cost is still considerable
>
> We agree the training cost is considerable. The inference speed is more than one order of magnitude faster than existing state-of-the-art, but can potentially benefit from further improved basis set, architecture, and other factors. We are excited to explore the directions proposed in the discussion section to further accelerate model training and inference in future works.
>
> > You mentioned in the text that if you use a different backbone other than eSCN, the performance is worse. It would be great if you could share some numbers to back up this statement.
>
> Thank you for the constructive feedback. We provide the performance metrics for a tensor field network backbone used in Charge3Net [1] and MACE [2], with no virtual nodes and $\beta = 2.0$. We compare these models to the small eSCN model with no virtual nodes and $\beta = 2.0$, so the basis set expressive power is the same. The hyperparameters for the baseline models are adopted from previous works and are included in the rebuttal PDF.
>
> | Model | NMAPE | Mol. per Min. |
> |-----------------|---------------|---------------|
> | Charge3Net | 4.79 ± 0.026 | 505.10 |
> | MACE | 5.14 ± 0.024 | 660.25|
> | eSCN | 0.504 ± 0.001 | 675.47 |
>
> We conclude the performance of the eSCN model is significantly better. We are also submitting our code, which can be used to reproduce all results in the paper, for peer review. Following NeurIPS 2024 instructions, we have sent an official comment to the AC that includes an anonymous link to the code base.
>
> We look forward to further discussions if you have additional questions or suggestions.
>
> [1] Koker, Teddy, et al. "Higher-order equivariant neural networks for charge density prediction in materials." npj Computational Materials 10.1 (2024): 161.
>
> [2] Batatia, Ilyes, et al. "MACE: Higher order equivariant message passing neural networks for fast and accurate force fields." Advances in Neural Information Processing Systems 35 (2022): 11423-11436.

---

> > ### Comment · Reviewer_paLd · 2024-08-08
> >
> > Do you have some qualitative, rough idea on why switching form eSCN to another backbone would cause such a dramatic drop in the performance?
> >
> > Other than that, all my questions are sufficiently addressed. Thank you again for your rebuttal.

---

> ### Author Response · Authors · 2024-08-08
> **Thank you for your response**
>
> Thank you for your response! A key distinction between eSCN and alternative backbones is eSCN uses point-wise spherical non-linearities. While all models use irreducible representations of SO(3) (irreps) for the internal representation of atomic features, to ensure equivariance, tensor field network and MACE can only apply non-linearities over the scalar features of the irreps. That is, only a small portion of the irreps features can be processed with non-linearities.
>
> On the other hand, the eSCN convolution maps the irreps to an equivalent spherical function and then apply point-wise spherical non-linearities, then maps the spherical functions back to irreps when the convolution finishes. This allows non-linearities over higher order tensor features while preserving equivariance. We hypothesize non-linearities over higher-order features makes eSCN a powerful architecture for the charge density prediction task. We believe our task is very challenging, and requires very expressive networks because the rich charge density information needs to be compressed into a rather compact set of basis set coefficients -- a highly complex mapping. We believe this task requires more expressive model than ML potentials (which only needs to learn a scalar energy and vector forces on the atoms), or probe-based charge density prediction (which doesn't have to compress the charge density to highly compact representations).
>
> Thank you again for your time and considerations. We are happy to discuss further if you have additional feedback.

---

### Author Rebuttal · Authors · 2024-08-06

Dear Area Chairs and Reviewers,

Thank you for your time and consideration in reviewing our paper. Following the suggestions from the reviewers, here we summarize our responses and several improvements we aim to make in the next version.

- We report the performance metrics of alternative model backbone architectures to support our choice of eSCN as our main backbone architecture.
- We elaborate on the significance and importance of learning to predict charge density in terms of downstream applications, ML applications, and novel insights of our paper. In particular, we emphasize the importance of efficiency in unblocking future research in this topic.
- We conduct additional experiments on the MD dataset [1,2] used in [3] (detailed in response to reviewer SS9k) and will add the new experiment results to the final version.
- We report the inference time scaling with regard to molecular size results to verify the theoretical linear scaling of our proposed method.
- We include more detailed descriptions of our design choices and model.
- We clarify the difference between charge density prediction models and ML potentials.
- We submit our code through an anonymous link for peer review. Following the NeurIPS 2024 guideline, we sent an official comment containing the link to the AC.

Our rebuttal PDF contains the following:

- A figure that demonstrates the inference time scaling with regard to molecular size.
- Pseudo code for the charge density prediction procedure of our proposed method.
- Hyperparameters for alternative model backbone architectures.

Thank you again for your feedback which greatly helped us improve the clarity and thoroughness of our paper. We look forward to further discussions if you have additional questions or suggestions.

[1] Bogojeski, Mihail, et al. "Quantum chemical accuracy from density functional approximations via machine learning." Nature communications 11.1 (2020): 5223.

[2] Brockherde, Felix, et al. "Bypassing the Kohn-Sham equations with machine learning." Nature communications 8.1 (2017): 872.

[3] Cheng, Chaoran, and Jian Peng. "Equivariant neural operator learning with graphon convolution." Advances in Neural Information Processing Systems 36 (2024).

---

### Decision · Program_Chairs · 2024-09-25

**Decision:**

Accept (poster)

**Comment:**

Despite the range of numerical scores on this paper, the text of the reviews is essentially unanimous, and in discussion, all reviewers converged on poster presentation for this paper.

Reviewers have suggested several areas for improvement and the rebuttal includes additional experimentation which reinforces the paper's contribution within the domain of charge density prediction.

A key question of reviewers and AC in deciding between poster and spotlight presentation was generalizability of the paper's contributions to a wider machine learning audience.  As the authors point out in the rebuttal, the model has some wider application ("to any geometric deep learning backbone models that use irreps (irreducible representations of SO(3)) for feature representations, which is common in modern equivariant architectures").  This does include applications outside charge density prediction, but remains a relatively small subset of wider ML research.